# Physical Forces in Glioblastoma Migration: A Systematic Review

**DOI:** 10.3390/ijms23074055

**Published:** 2022-04-06

**Authors:** Audrey Grossen, Kyle Smith, Nangorgo Coulibaly, Benjamin Arbuckle, Alexander Evans, Stefan Wilhelm, Kenneth Jones, Ian Dunn, Rheal Towner, Dee Wu, Young-Tae Kim, James Battiste

**Affiliations:** 1Department of Neurosurgery, University of Oklahoma Health Sciences Center, Oklahoma City, OK 73104, USA; audrey-grossen@ouhsc.edu (A.G.); kyle-d-smith@ouhsc.edu (K.S.); nangorgo-coulibaly@ouhsc.edu (N.C.); benjamin-arbuckle@ouhsc.edu (B.A.); alexander-evans@ouhsc.edu (A.E.); ian-dunn@ouhsc.edu (I.D.); rheal-towner@omrf.org (R.T.); 2Stephenson School of Biomedical Engineering, University of Oklahoma, Norman, OK 73019, USA; stefan.wilhelm@ou.edu; 3Stephenson Cancer Center, The University of Oklahoma Health Science Center, Oklahoma City, OK 73104, USA; 4Institute for Biomedical Engineering, Science, and Technology (IBEST), University of Oklahoma, Norman, OK 73019, USA; 5Department of Cell Biology, University of Oklahoma Health Sciences Center, Oklahoma City, OK 73104, USA; ken-jones@ouhsc.edu; 6Department of Radiological Sciences, University of Oklahoma Health Sciences Center, Oklahoma City, OK 73104, USA; dee-wu@ouhsc.edu; 7Department of Bioengineering, The University of Texas at Arlington, Arlington, TX 76019, USA; ykim@uta.edu; 8Department of Urology, The UT Southwestern Medical Center, Dallas, TX 75390, USA

**Keywords:** glioblastoma, chemoresistance, physical forces, tumor microenvironment

## Abstract

The invasive capabilities of glioblastoma (GBM) define the cancer’s aggressiveness, treatment resistance, and overall mortality. The tumor microenvironment influences the molecular behavior of cells, both epigenetically and genetically. Current forces being studied include properties of the extracellular matrix (ECM), such as stiffness and “sensing” capabilities. There is currently limited data on the physical forces in GBM—both relating to how they influence their environment and how their environment influences them. This review outlines the advances that have been made in the field. It is our hope that further investigation of the physical forces involved in GBM will highlight new therapeutic options and increase patient survival. A search of the PubMed database was conducted through to 23 March 2022 with the following search terms: (glioblastoma) AND (physical forces OR pressure OR shear forces OR compression OR tension OR torsion) AND (migration OR invasion). Our review yielded 11 external/applied/mechanical forces and 2 tumor microenvironment (TME) forces that affect the ability of GBM to locally migrate and invade. Both external forces and forces within the tumor microenvironment have been implicated in GBM migration, invasion, and treatment resistance. We endorse further research in this area to target the physical forces affecting the migration and invasion of GBM.

## 1. Introduction

The mechanobiology of brain tumors is a vast and essential part of understanding their growth, progression, and chemoresistance [1]. Over the last two decades, there has been continuous study and development of the molecular biological underpinnings of glioblastoma (GBM), but with little focus on the relationship between physical forces and migration. In GBM, it is known that certain molecular aberrations exhibit more aggressive migratory patterns. Classic glioma biomarkers include isocitrate dehydrogenase (IDH) mutation, 6-methylguanine–DNA methyltransferase (MGMT) modification, and the deletion of 1p19q, which are hallmarks of the aforementioned molecular profiling that lead to the backbone of GBM classification, nomenclature, and scientific research, but they are more associated with DNA repair than invasive characteristics. Emerging research focuses on the complex contributions of the physical forces to cancer aggression, invasion, and migration.

Emerging data indicates that tissue invasion increases GBM aggressiveness, chemoresistance, and overall mortality [2]. Brain invasion is associated both with poor prognosis and a median survival of under one year for a majority of patients. This invasion is often accompanied by neurologic dysfunction leading to reduced quality of life. A myriad of potential targets are emerging for further research into the invasion and migration of GBM. The tumor microenvironment influences the molecular behavior of cells, inducing mutations. Current forces being studied include the properties of the extracellular matrix (ECM), such as stiffness and “sensing” capabilities [3].

The brain presents unique challenges when it is affected directly by cancer, as it is confined within the rigid skull. This poses questions with regard to how increased edema, producing elevated intracranial pressure (ICP), compression, tension, and other mechanical forces, affects GBM. Additionally, the brain also uniquely has the blood–brain barrier (BBB) and is an “immunologically privileged” anatomical location. The effect that the input of these physical forces has on the hallmark macrophage/microglial infiltration in GBM is not well understood. However, it is known that these tumors disrupt the BBB integrity and have the potential to alter the ECM [4]. However, it is also known that in diffuse GBM, the BBB remains essentially intact, which has reduced some therapeutic advances [5].

There is currently limited data on the physical forces in GBM—regarding both how they influence their environment and how their environment influences them. This review outlines the advances that have been made in the field. It is our hope that further investigation of the physical forces in GBM will highlight new therapeutic options and increase patient survival.

## 2. Results

Our search yielded 98 unique results. Of these, 30 were included in analysis (Figure 1). These studies discussed 11 external/applied/mechanical forces and 2 tumor microenvironment (TME) forces that affect the ability of GBM to locally migrate and invade (Table 1 and Table 2).

**Figure 1 ijms-23-04055-f001:**
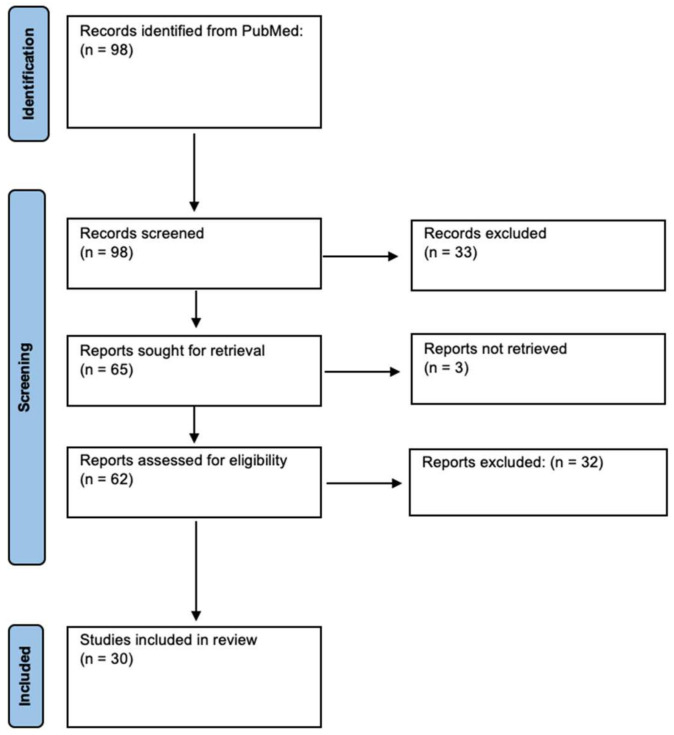
Identification of Studies via PRISMA Guidelines.

**Table 1 ijms-23-04055-t001:** External/Applied/Mechanic Forces in GBM.

	*Author*	*Stress Marker*	*Study Design*	*Effect on GBM*
** *Stiffness* **	Chen et al. [6]	Piezo/PIEZO1	Drosophilia glioma model in vivo; mice xenograft experiments; RNA sequencing of two human GBM stem cell lines (G508 and G532)	Regulator of mitosis and tissue stiffness through activation of integrin-FAK signaling; correlated with GBM aggressiveness and decreased survival
	Miroshnikova et al. [7]	Tenascin C	Patient-derived samples; mouse model	ECM stiffness represses miR-203 expression which activates HIF1α-dependent TNC deposition, which may induce aggressiveness and lead to recurrence
	Sen et al. [8]	Talin-1	U373 MG human glioma cells	Involved in mechanical rigidity sensing; transmits signals from the ECM to the cytoskeleton through interplay of integrins and actin
	Khan et al. [9]	N/A	CD 133+ GBM cells	Actively migrating GBMs exhibit higher elastic stiffness at the front end, facilitating traction needed for forward movement through an anchoring effect
** *Tensile Force* **	Barnes et al. [10]	Tension (tenescin)	Patient-derived samples; mouse model	Tension-mediated glycocalyx–integrin feedback loop which promotes mesenchymal characteritistics
	Shen et al. [2]	Yes-associated protein (YAP)	G55 GBM cells	Re-localization of YAP to the cell nucleus indicates a higher degree of cytoskeletal tension during migration of GBM cells in a physically confined environment
** *Traction* **	de Semir et al. [11]	Pleckstrin homology domain-interacting protein (PHIP)	In vitro and in vivo murine model of U-251 GBM cell lines	Plays a role in activating the actin cytoskeleton, focal adhesion dynamics, migration, and invasion
	Gordon et al. [12]	Latex beads displacement and cell line volumetric growth	In vitro using human U87MGmEGFR GBM cell line	Demonstrated that tumor cells will grow towards the path of least resistance through traction-mediated forces
** *Drag Force* **	Agosti et al. [13]	N/A	U87 GBM cell lines	During proliferation, GBM aggregation is enabled when the adhesive force between cells is of the same magnitude of the drag forces of cells as they expand
** *Compression* **	Voutouri et al. [14]	Vessel option	Mathematical model	Compression led to hypoxia and resultant angiogenesis
	Calhoun et al. [15]	miR548 family	LN229 and U251 GBM cell lines; pathway analysis	Increased migration and decreased proliferation, characteristics associated with tumor aggressiveness
	Demou et al. [16]	Caveolin-1, integrin-β1, Rac1	U87 and HGL21 GBM cells	Cell deformation/compression leads to downregulation of E-cadherin (CDH1) and PECAM-1 (CD31) and overexpression of PTEN and Rac1; resultant decrease in cell adhesion and increased migration
** *Adhesion* **	Morjen et al. [17]	Kunitz-typeprotease inhibitor (PIVL)	In vitro using U87 cell lines; in vivo mouse model	Disrupted GBM migration, invasion, and adhesion through inhibition of integrin
	Yao et al. [18]	P311/PTZ17	In vivo mouse model	Rho GTPase-mediated promotion of migration of epidermal stem cells
** *Hydrostatic Pressure* **	Claus et al. [19]	N/A	Case report	Increased CSF protein concentration caused increased ICP and patient deterioration
	Takara et al. [20]	N/A	Case report	Increased CSF protein concentration led to hydrostatic pressure build up
	Zoi et al.	Polycystin-1 (PC1)	T98G GBM cells subjected to coninuous hydrostatic pressure and/or PC1 blockade	Hydrostatic pressure inhibited proliferation and migration of GBM cells. PC1 had the opposite effect
** *Magnetic Force* **	Perez et al. [21]	N/A	(U87) tumor spheroid aggregation methodology based on magnetic cell labeling; spheroid cell invasion w/ Matrigel	Magnetic properties of the spheroids allow for determination of surface tension
	Chen et al. [22]	Hexagonal superparamagnetic cones	U-343 GBM cell lines	Magnetic field gradient decreased cell growth and migration
** *Osmotic Pressure* **	Catacuzzeno et al. [23]	Swelling-activated chloride currents	In vitro using GL-15 GBM cells	Channel activation included shape and volume changes, allowing migration and invasion
	Pu et al. [24]	Caveolin-1, CAVIN1; uPA and MMPs; AQP1	U87, U118, and U251 GBM cell lines	Play a role in the response to increased pressure and GBM invasion
	Pu et al. [25]	Snail-1, Snail-2, N-cadherin, Twist, and vimentin	GBM cell lines U87 and U251; patient-derived neural oncospheres	EMT and invasion through production of matrix proteases as a response to osmotic/hydrostatic pressure
** *Shear Stress* **	Rezk et al. [26]	Nestin and vimentin; actin filaments, vinculin, paxillin, and FAK	Patient-derived samples	Increased migration and proliferation
** *Solid Stress* **	Ciarletta et al. [27]	N/A	Theoretical calculation of buckling instability from solid stresses	Residual stresses promote buckling instability and promote tumor invasion
	Stylianopoulos et al. [28]	Collagen, hyaluronan	Mathematical model	Increased perfusion of tumors led to improved oxygenation and drug delivery

N/A: Not applicable.

**Table 2 ijms-23-04055-t002:** Tumor Microenvironment Forces in GBM.

	*Author*	*Stress Marker*	*Study Design*	*Effect on GBM*
** *Cellular Volume* **	Fischer et al. [29]	HAMLET	In vitro using non-transformed human astrocytes CC-2565; in vivo animal models using human GBM xenografted rat models	HAMLET selectively induced GBM apoptosis in rat xenograft models via activation of programmed cell death. HAMLET did not interact with healthy tissue and extended survival by relieving GBM pressure symptoms via volume reduction
	Sforna et al. [30]	Swelling-activated chloride currents	In vitro using GL-15, U87MG, and U251 cells lines	Acute and cyclic hypoxic conditions (either blood flow interuptions) may enable GBM cells to upregulate I(Cl,swell) conditions, which regulate the cellular volume and prevent cellular death
** *Intracranial Pressure* **	Chida et al. [31]	N/A	Case report	Increased high CSF protein and pressure hypothesized to cause aggressive phenogype
	Rifikinson-Mann et al. [32]	N/A	Case series of hydrocephalus associated with intramedullary spinal GBM	Malignant tumors were associated with tumor extension and ventriculomegaly
	Yoo et al. [33]	Hyaluronic acid	U87MG, U373MG, and U251MG glioma cells; transwell assay	In response to radiation, HA production was increased in GBM cells by HA synthase-2 (HAS2), which was transcriptionally upregulated by NF-ĸB. Notably, NF-ĸB was persistently activated by an IL-1α-feedback loop, making HA abundant in tumor microenvironment after radiation

N/A: Not applicable.

## 3. Discussion

### 3.1. External/Applied/Mechanical Forces

Physical forces have varying impacts depending on many factors, including the rigidity of the object, the composition of the object, and the geometry of the object. At the molecular and cellular level, the applied physics cause cellular responses depending on the nature of the force and the intensity of its application. The brain has compensatory tools available, with the ability to adjust to changes when physical forces are applied. Blood cells traversing through capillaries have been thoroughly studied, with their unique geometry and internal structure aiding in the delivery of oxygen to tissues while maintaining structural integrity for up to 120 days. Some cellular responses to external forces are based on the physical characteristics of the cell, such as the blood cell. Other responses to external stimuli can produce a biomechanical response, such as mechanoreceptors opening their ion channels to pressure on peripheral neurons, or a regulatory response, such as the upregulation of angiogenesis recruitment chemicals, such as VEGF [34]. In tissues, the physiologic response is a directed transmission of applied forces to invoke a downstream function.

#### 3.1.1. Tensile Force and Stiffness

Tensile forces are physical forces that induce a net strain on an object. Tensile strength is a key mechanical property of materials, indicating their strength and elasticity. The differences in tension in an intra- or extracellular matrix could induce adaptations and changes in growth patterns and behaviors. In GBM, the tension force exerted on the tumor cell has a unique impact on the growth and aggression of the tumor (Figure 2). GBM increased the production of tenascin secreted into the extracellular matrix, increasing the overall extracellular environment’s stiffness, potentiating growth, survival, and invasion [7,35].

Increased environmental stiffness and bulkiness, through the upregulation of both tenascin and glycoproteins, were also associated with a self-reinforcing mesenchymal-to-epithelial transition [10]. Barnes et al. further described how GBM transition and growth through the increased bulkiness of the glycocalyx is mediated via an integrin mechano-signaling-linked regulatory circuit, a mutant V737N integrin β1 that enhances FAK activity, causing a self-perpetuating cycle of increased ECM glycocalyx bulkiness and tenascin production. A proposed mechanism for the sensation of this increased tension and bulkiness is through a mechano-sensitive ion channel PIEZO1, localized at adhesion points [6]. The channel then activates integrin-FAK signaling, which reinforces tissue stiffening and promotes tumor aggression. Another focal adhesion protein, talin-1, has been implicated in the aggression and migration of GBM U373 cell lines, though their role in other GBM cell lines and in vivo tumors remains under investigation [8]. Differences in the increased ECM stress were found to be dependent on the active migration status of the GBM. When migrating, the redistribution of the actin and myosin is towards the migratory front ends, using the increased tension and stiffness to enhance the tumor anchoring effect [9]. There was greater energy distribution while in the migratory state than in the non-migratory state, decreasing the frictional resistance to the GBM migration [9].

Tension adaptation is a physiological necessity for healthy tissue adaptation. However, excess and prolonged extraneous tension can exacerbate disease states. In cirrhotic livers, tension was linked to the development of cancer via the upregulation of mRNAs [36]. Elasticity was used to aid in the screening of breast cancer with increased efficacy over standard screening methods [37]. A similar method of determining tissue elasticity via ultrasound was used in pancreatic patients with only modest results [38].

#### 3.1.2. Compressive Force

Compressive forces are physical forces that have a net inward vector on an object. Compressive forces in GBM produce a range of physiological responses from the tumor, leading to the induction of migration, the upregulation of epigenetic signals, and the formation of new blood vessels [14,15,16]. In U87 and HGL21 cell lines, compression strain induced the downregulation of E-cadherin (CDH1) and PECAM-1 (CD31) and the upregulation of PTEN and Rac1. The downstream effects were a decrease in cell adhesion and increased cellular migration, respectively [16]. In the Ln229 and U251 GBM lines, mechanical compression led to miR548 family induction of epigenetic signaling. The induced signaling was correlated to cell elongation, increased migration, decreased proliferation, and increased tumor aggression characteristics [15]. The induced mechanical compression in the study by Calhoun et al. was also associated with decreased survival and increased therapy resistance, which they suggest may be due to the enhanced migration and escape mechanisms from focal surgical or chemotherapeutic treatment [15]. A mathematical model determined that compression induced vessel cooption, leading to hypoxia and new vessel formation via angiogenesis [14]. Three different compressive forces studied had overlapping, though different, physiologic responses according to their GBM models. The aggressiveness of GBMs often induces increased pressures and compressions on their environment due to the limited volumetric capacity in which they grow (Figure 3). The compressive forces may also impact the tumor directly: as the total environmental pressures increase with the tumor’s growth, the tumor may eventually collapse in on itself [27]. The tumor collapse may be associated with tumor invasion. This induction of compressive forces potentiates the survival of the affected cells, inducing aggression in uninhibited malignancies [27].

The stimulatory response to compressive forces is not unique to GBMs. In HeLa cells, compression upregulated autophagy and promoted paxillin turnover and MMP-2 secretion, all of which induced cellular migration [39]. Epithelial ovarian cancer in vitro studies describe how compression-altered genes relate to the epithelial–mesenchymal transition [40]. Acute compressive forces were applied to cells and tissues suspended in a three-dimensional construct and were found to elevate RHOA-GTP levels and regulatory myosin phosphorylation with actomyosin contractility via ROCK. This led to increased expression of EMT regulatory and cellular proliferation [41]. The range of compensatory responses to a hyper-pressurized environment underscores the homeostatic responses of normal tissues and potential malignant responses.

Increased compressive forces or solid stresses in tumors can also inhibit drug delivery to tumors. Enhanced solid stresses have been linked to the devascularization of solid tumors and their environment [42]. The solid stresses not only increase tumor survivability and aggression, but also its resistance to drug delivery. Opening and perfusing blood vessels can then lead to the issue of delivering nutrients to the tumor if not appropriately paired with cytotoxic treatments. Thus, combined treatments of improving tissue oxygenation and drug delivery must be considered when normalizing vasculature [28].

#### 3.1.3. Adhesive, Traction, and Drag Forces

The interplay of adhesive, traction, and drag forces is illustrated in Figure 4. Adhesion generally refers to the attractive forces between two different materials or substances. Electrostatic forces, protein–protein interactions, or mechanical forces can all contribute to adhesion. In tumors, adhesive proteins are heterogenous, making therapeutic targeting limited and mostly ineffective. By sampling different tumor locations, lower adhesive forces were found at the leading migratory edges than at the anchoring edges [26]. The heterogeneity in adhesive profiles implicates a genetic or epigenetic heterogeneity based on the GBM cell location and environment. Kunitz-type protease inhibitors were used in GBM U87 cell lines and PIVL was able to disrupt U87 migration, invasion, and adhesion via the disruption of cellular surface fibrin and fibrinogen of the extracellular matrix [17]. In vivo studies are needed to replicate this disruption, but this suggests that anchoring may be the key to the survival of GBM.

Traction forces are the forces involved in producing movement between two surfaces. Sufficient friction between the surfaces is necessary to prevent slipping, but it cannot be so great that the object remains adhered tangentially. Traction forces are closely related to adhesive forces, especially biologically. In GBM U251 cell lines, the pleckstrin homology domain-interacting protein drove motility and invasion by acting on the force transduction layer of the focal adhesion complex and regulating the actin cytoskeleton, focal adhesion dynamics, and tumor cell motility [11]. GBM U87 cell lines induced traction forces on nearby ECM locations and pressure on distant ECM locations, enabling a ‘mapping’ of the path of least resistance of tumor growth [12].

Adhesive and traction forces have been characterized as inducing or influencing other cancers. Adhesive and traction states have been well studied, especially in breast cancers [43,44,45,46,47,48,49,50,51,52,53,54,55,56,57,58,59]. Lung and prostate cancers have also been explored with a similar approach [59,60,61,62]. These trials and methods should be used in future research into understanding the biomechanics and changes in GBM, especially in the modeling of traction forces and the growth mechanics in developing better predictive models and therapeutic interventions.

Drag forces are the proportional force against an object within a flow. In GBMs, this then refers to the force gradient surrounding the migratory tumor when anchored to the surrounding tissues. While not as prominent of a force compared to pressure or tensile forces in the CSF, the characterization of increased aggregation proportionally to the drag forces was found in GBM U87 cell lines [13]. The environmental drag enabled the self-organization and aggregation of the cellular network to the same magnitude as the surrounding drag.

Perhaps more than their use in causing cancerous changes, drag forces have been used in combination with microparticles to determine information on fluid microenvironments and surrounding cells in cancerous settings. Adhesion strength and mechanics were determined via acoustic stimulated drag forces that sheared strength-specific breast cancer cells from the surrounding medium [63]. The disruption could be linked to the aggressiveness of the cells and could be therapeutically informative of the nature of different breast cancers. A microfluidic device was developed that was able to target hyper-aggressive cancer cells on the basis of their diaphoretic signature and Stokes drag force. Optically induced electro-kinetic microfluidic devices were used to determine leukemic properties in vivo via the analysis of cellular drag force [64].

#### 3.1.4. Hydrostatic and Osmotic Pressure

Hydrostatic pressure is the outward pressure exhibited by fluids in proportion to external pressures applied to the fluids. The spinal canal and cranial sinuses are a fixed volume surrounded by incompressible materials of the brain and spinal cord. In healthy individuals, there is a balance of forces between the cerebral spinal fluid, the space occupied by the neural matter, and the vascular supply. The exchange of fluid from the intracerebral and intraspinal to the vascular supply is limited due to the robustness of the blood–brain barrier. As such, when masses develop in the spine or the brain, the total volume is decreased. Because of the inability to readily shift fluid or change the volume of the cranial and spinal spaces, the pressure will build throughout the CNS. Both the mass effects of the tumor and increased pressure will produce many of the initial symptoms of GBMs and are associated with a decreased quality of life and survival [31].

Oncotic pressure is the pressure effects due to the protein concentration gradient. Shifts in fluids are induced by the up- and down-productions of proteins, where fluids will follow increased pressures, especially in fenestrated vessels. A physiologically normal gradient exists to enhance the proper flow of fluids; in diseased states, the gradient is disturbed to either pull or release fluid outside of the normal boundaries. As such, the oncotic pulling of fluid into the CNS spaces due to the increased production of proteins will increase the overall CSF pressure, contributing to the previously discussed hydrostatic pressure effects (Figure 5). [19,20,31,65].

The pressurized environment contributes to GBM tumorgenicity. As the tumor grows, the environment becomes more hypoxic as the body attempts to combat the high nutrient consumption of the tumors (see hypoxia discussion). The hypoxia and fluid pressures induce adaptive measures, as tested in GL-15 GBM cell lines [23]. The cells upregulate swelling-activated chloride currents, enabling cytoskeleton remodeling and volume and shape changes, leading to enhanced migration and invasion. In the U87, U118, and U251 GBM cell lines, enhanced pressure proportionally upregulated caveola-forming proteins in addition to AQP1, contributing to the invasiveness of the cell lines [24]. Enhanced invasiveness in response to pressure was also found in the U87 and U251 GBM cell lines, as the stress affects cell processes, including signal transduction and overall regulatory processes via Snail-1, Snail-2, N-cadherin, Twist, and vimentin upregulation [25].

A wide range of techniques have been employed to study oncotic and hydrostatic pressures in cancer. Different cancer studies have included adenocarcinoma [66], breast [67], colorectal [68], esophageal [69], head and neck [70], lung [71], melanoma [68], ovarian [72], prostate [73], and skin [74]. The variations in hyper- and hypo-osmotic states contributing to the tumorgenicity of the different cancers demonstrates the importance of regulating homeostasis.

### 3.2. Tumor Microenvironment (TME)

#### 3.2.1. Intracranial Pressure

Increased intracranial pressure is defined as an elevated pressure within the skull. This is a common clinical problem encountered in patients with brain tumors. However, the effects of increased pressure on tumor cell migration are not fully understood. The increased pressure is due to compounded forces that are applied to the brain and can be caused by additional fluid, or the growth of a brain tumor that applies more physical forces than normal on the brain all within the rigid space contained by the skull (Figure 6). Normal supine intracranial pressure is between 7 and 15 mmHg. One case report of a woman diagnosed with GBM in the postpartum period without signs of myelopathy presented with ~20.6 mmHg opening pressure on a lumbar puncture. It was assumed that the GBM primarily grew in the cervical cord and metastasized into the intracranial subarachnoid space [31].

In another study, 171 patients with intramedullary spinal cord tumors underwent surgical resection. Twenty patients had a malignant tumor, in which thirteen out of twenty cases were complicated by increased intracranial pressure and ventriculomegaly. Of the remaining 151 patients, an addition 12 developed systematic hydrocephalus. Increasing intracranial pressure generates a holocephalic compressive force that causes the compression of neoplastic and normal brain tissue, simultaneously creating a global effect on all tissues in the brain.

Hyaluronic acid (HA) is a component of brain outflow pathways that has been shown to regulate fluid movement [75]. The increased production of HA can potentially lead to increased ICP. Yoo et al. [33] described how HA production was increased in GBM cells following radiation. This mechanism included the upregulation of HA synthase-2 (HAS2) by NF-ĸB. Notably, NF-ĸB was persistently activated by an IL-1α-feedback loop, making HA abundant in the tumor microenvironment after radiation

#### 3.2.2. Cellular Volume

Cellular volume is defined by the amount of fluid (primarily water) contained within the cell. Because of osmosis, the cellular volume is usually determined by the cellular environment: hypertonic, isotonic, or hypotonic. GBM cells express abundant Cl channels whose activity supports cell volume and membrane potential changes (Figure 7). Similar to non-tumor tissues, Cl channels are modulated by hypoxia in GBM. Acute hypoxia increased the cell volume by 20%. However, when GBM cells are in a 30% hypertonic environment, they showed partial inhibition of the hypoxia-activated Cl current. I_Cl,swell_ was observed to mediate the regulatory volume decrease in GBM, and increase the hypoxia-induced necrotic death in GBM. As a result, cellular volume through Cl channels plays a role in the survival of GBM cells [30].

#### 3.2.3. Hydraulic Conductivity

Hydraulic conductivity refers to the ability to conduct water, or fluids in general. In tumors, blood perfusion is lower than in normal tissue due to the compression of the tumor mass, or due to a higher permeability of vessels [9,28]. Thus, tumors are considered to have a lower hydraulic conductivity than regular tissue. Depending on the cause of the low hydraulic conductivity, vascular normalization occurs because of a decrease in vascular permeability, or vascular decompression to alleviate forces in the tumor [9,28].

#### 3.2.4. Adhesion Protein Expression

Adhesion proteins are cell membrane proteins that participate in interactions between cells (Figure 8). PIVL, a serine proteinase inhibitor, presents as a monomeric polypeptide chain cross-linked by three disulfide linkages. PIVL has shown the ability to inhibit the adhesion, migration, motility, and invasion of GBM U87 cells. The anti-cancer effect of PIVL is attributed to its (41)RGN(43) motif [17]. Another protein, P311, has also been proven to play a key role in GBM invasion. In human epidermal cells, P311 significantly accelerated cell migration in vitro and enhanced Rho GTPases activity when highly expressed. A RhoA-specific inhibitor and Rac1 inhibitor could both be used to significantly suppress P311-induced human epidermal cells [18].

### 3.3. Major Molecular Mechanisms Associated with Physical Forces in GBM

We have reviewed 11 physical mechanisms involved in GBM aggression, recurrence, migration, and invasion and, in summary, we identified 34 influential molecules and pathways. These molecular influencers were elucidated primarily via the analysis of human cell lines, and included G508, U373 MG, CD133+ GBM cells, U251, U87MG, U87, LN229, HGL21, U343, GL15, U118, and CC2565. The U87 cell lines were used most frequently in the reviewed studies. Three molecular pathways (Piezo/PIEZO1 [5], tenascin C [6], and Talin-1 [7]) were found to be involved in altering GBM stiffness, one (tenascin [9]) was involved in tensile forces, one (PHIP [10]) was involved in traction, four (miR548 [14], caveolin-1, integrin-β1, and Rac1 [15]) were involved in compression, two (PIVL [16] and P311/PTZ17 [17]) was involved in adhesion, ten (swelling-activated chloride current [22], caveolin-1/CAVIN1, UPA, MMPS, AQP1 [23], Snail-1, Snail-2, N-cadherin, Twist, and vimentin [24]) were involved in changes to cellular osmotic pressure, seven (Nestin, vimentin, actin filaments, vinculin, paxillin, and FAK [25]) contributed to shear stress, two (collagen and hyaluronan [27]) contributed to solid stress, and two (HAMLET [28] and swelling-activated chloride currents [29]) were involved in cellular volume changes. Most of the contributing molecular changes were not directly overlapping, though several contributed to more than one physical force: tenascin contributed to both stiffness and tensile forces; swelling-activated chloride channels impacted both changes to cellular osmotic pressure and cellular volume changes; vimentin influenced osmotic pressure and shear stress; hyaluronan impacted solids. The role of multiple molecular pathways influencing single cellular functions highlights GBM redundancy and suggests that the physical forces associated with more molecular pathways are critical to GBM survival. Furthermore, the diversity and variability of molecular changes to GBMs are telling of GBM’s robustness and adaptability.

## 4. Material and Methods

### 4.1. Search Strategy

A systematic review of studies involving physical forces and their effect on GBM was performed using the Preferred Reporting Items for Systematic Reviews and Meta-Analyses (PRISMA) guidelines. A search of the PubMed database was conducted from 13 September 2021 to 23 March 2022 with the following search terms: (glioblastoma) AND (physical forces OR pressure OR shear forces OR compression OR tension OR torsion) AND (migration OR invasion). Articles were reviewed by one team member, and interpretation was verified by a second team member.

### 4.2. Eligibility Criteria

The following inclusion criteria were utilized: (1) studies must include a specified force in GBM, (2) this force must compose either part of the tumor microenvironment or physically act on GBM, and (3) original research was presented. Reviews, meta-analyses, commentary, letters to the editor, editorials, and articles not accessible in English were excluded.

### 4.3. Data Extraction

The following data points were extracted from each study: physical force, experimental methods, effect on the migration and/or invasion of GBM, impact on radio-chemotherapy resistance, and any relation to overall survival. Each data point was extracted, reviewed, and agreed upon by two reviewers.

## 5. Conclusions

Both external forces and forces within the tumor microenvironment are involved in GBM migration, invasion, and treatment resistance. We endorse further research in this area to target the physical forces and the signaling pathways that transduce their effect on cells. The therapeutic inhibition of the migration and invasion of GBM could represent new therapeutic avenues. Current research opportunities in this field include the establishment of better GBM research models and drug screening systems that incorporate physical forces into the assessment of tumor cell biology and drug efficacy.

## Figures and Tables

**Figure 2 ijms-23-04055-f002:**
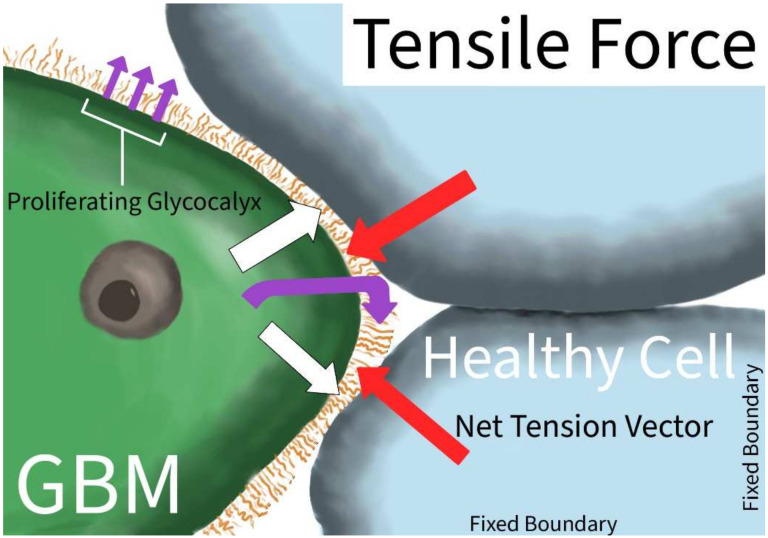
Tensile Force in GBM. Tension vector (white arrow) applied to GBM cell exerts force on the GBM cell membrane, and in response, the GBM produces an extracellular glycocalyx matrix (purple curved arrow) leading to matrix growth (purple straight arrow); the glycocalyx matrix can pull on the surrounding healthy tissue, inducing net tensile force at the leading borders of the GBM (red arrow).

**Figure 3 ijms-23-04055-f003:**
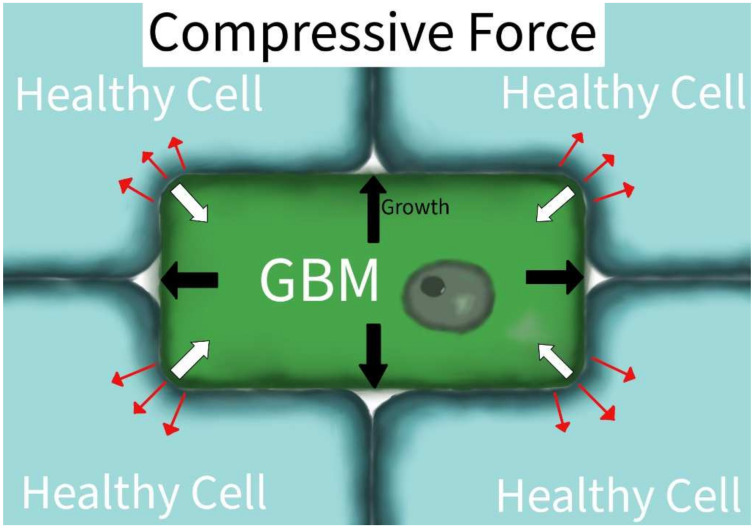
Compressive Force in GBM. The external environment will exert compressive forces on the GBM cell (white arrow vectors). When GBM grows in a fixed volume or is surrounded by immobile tissue (black arrows), it will also exert compressive forces on the surrounding tissue structure (red arrows).

**Figure 4 ijms-23-04055-f004:**
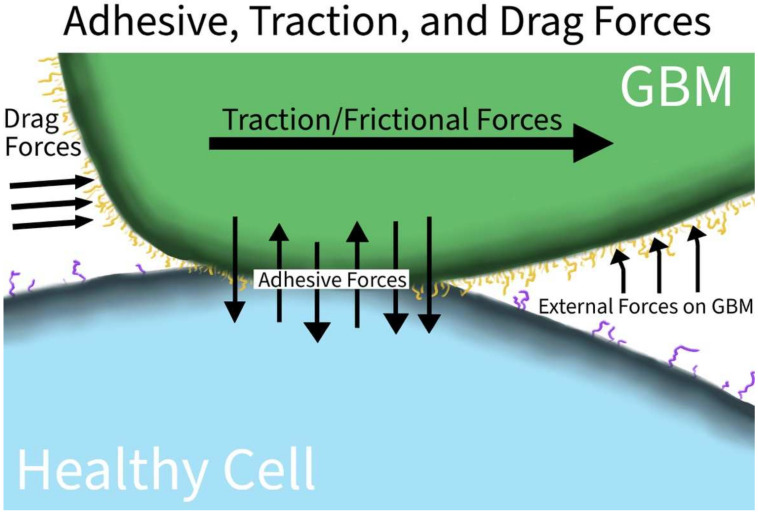
Adhesive, Traction, and Drag Forces in GBM. GBM upregulates various surface proteins, enabling it to adhere to the surfaces of healthy tissue; the increased adherence also helps GBM to resist the physical forces of other tissues or fluids via increased traction forces directly at the surface interface and drag forces at the free margins of the cancer.

**Figure 5 ijms-23-04055-f005:**
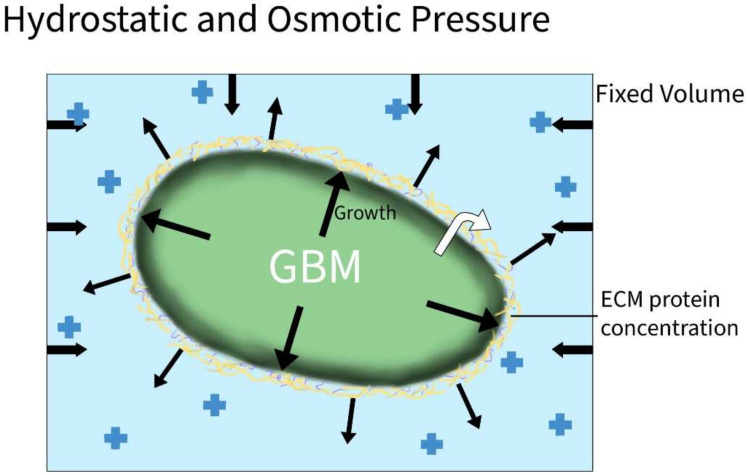
Hydrostatic and Osmotic Pressure in GBM. As GBM grows and produces excess proteins (black cellular arrows and yellow margins) in the fixed craniospinal volume, both hydrostatic and osmotic pressures (black arrows at edges) will build as the intercranial fluid and extracellular protein concentrations continue to increase.

**Figure 6 ijms-23-04055-f006:**
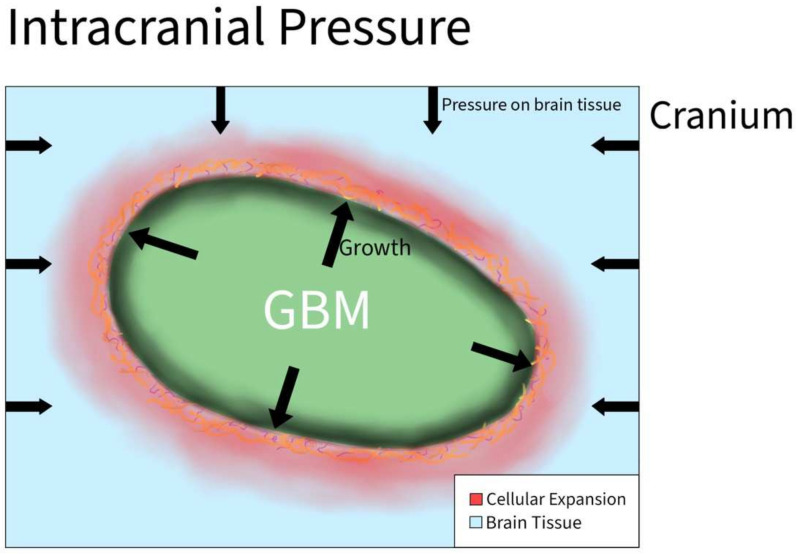
Intracranial Pressure in GBM. As the tumor grows rapidly inside the brain, the overall size of the brain increases and causes tissue to start pressing against the cranium. As a result, the cranium exerts compressive forces back on the brain that result in an increase in intracranial pressure.

**Figure 7 ijms-23-04055-f007:**
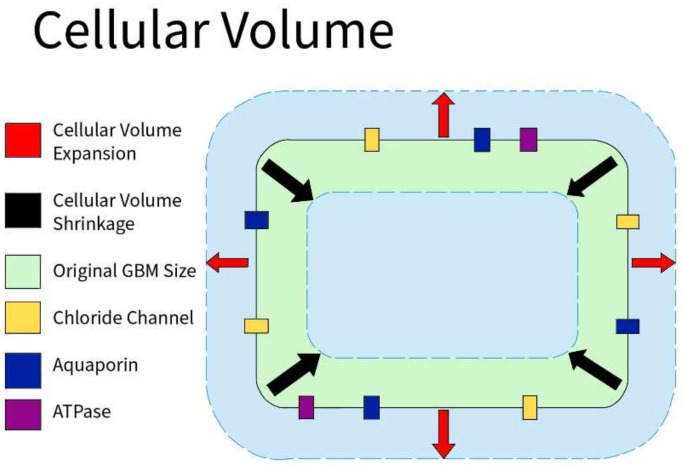
Cellular Volume in GBM. GBM cells express an abundance of chloride ion channels. Along with aquaporin channels and various ATPases, those channels allow the cells to shrink or swell depending on the environment to aid in the survival of GBM cells.

**Figure 8 ijms-23-04055-f008:**
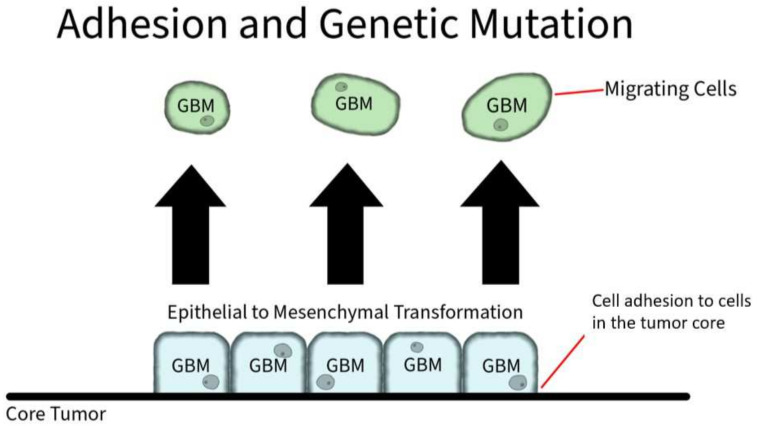
Adhesion and Genetic Mutation in GBM. Cellular membrane proteins play a role in individual GBM cell adhesion to the core tumor. However, through genetic mutation, GBM cells can induce an overexpression of hyaluronic acid, which serves as a ligand for CD-44 receptors. The CD-44 receptors activate SRC complexes that induces a shift to mesenchymal shift in GBM.

## Data Availability

Search conducted via PubMed database.

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
