# Peer review of "Physical Forces in Glioblastoma Migration: A Systematic Review"

_ijms, 2022, doi:10.3390/ijms23074055_

Round 1

Reviewer 1 Report

Dear authors,

You have done a careful review and extensive review, but to point of view thee are several aspects that must be clarified. These are the following:

  1. The term glioblastoma should be used instead of glioblastoma multiforme.
  2. Line 58: How can chemoresistance be defined by invasion capability?
  3. Line 66: According to the literature hypoxia is not a physical force.
  4. Line 69: Edema is the cause of increased ICP.
  5. Lines 126-127: Cells don’t have blood flow.
  6. Lines 130-131: A reference is needed.
  7. Lines 161-162: A reference is needed.
  8. Lines 188-190: How was the resistance to therapy proved?
  9. Lines 198-201: The mechanisms of this statement must be explained and references presented.
  10. Lines 327-330: Which are the therapies that target osmotic and pressures imbalances?
  11. Chapter 4.2.1: Hypoxia isn’t a physical force.
  12. Chapter 4.2.2: Increased intracranial pressure is a clinical problem and is common complication of brain malignancies.
  13. Line 382: Is the term postpartum correct?
  14. Chapter 425: Genetic mutations aren’t a physical forces

Author Response

Dear authors,

You have done a careful review and extensive review, but to point of view there are several aspects that must be clarified. These are the following:

  1. The term glioblastoma should be used instead of glioblastoma multiforme.

We thank this reviewer for their comments. We have removed “multiforme” from the manuscript.

  1. Line 58: How can chemoresistance be defined by invasion capability?

We have rephrased this sentence and have provided a reference for more clarity.

“Emerging data indicates that tissue invasion increases GBM aggressiveness, chemoresistance and overall mortality.2

  1. Line 66: According to the literature hypoxia is not a physical force.

We have removed hypoxia as a physical force discussed within the paper.

  1. Line 69: Edema is the cause of increased ICP.

We have reworded this phrase:

“The brain presents unique challenges when it is affected directly by cancer as it is confined within the rigid skull. This poses questions with regard to how increased edema, producing elevated intracranial pressure (ICP), compression, tension, and other mechanical forces affect GBM.”

  1. Lines 126-127: Cells don’t have blood flow.

We have stated that that blood cells flow through capillaries, not that blood cells have blood flow.

  1. Lines 130-131: A reference is needed.

We have provided a reference.

  1. Lines 161-162: A reference is needed.

We have provided a reference.

  1. Lines 188-190: How was the resistance to therapy proved?

We have added to this sentence and provided the appropriate citation:

“The induced mechanical compression in the study by Calhoun et. al. was also associated with decreased survival and increased therapy resistance, which they suggest may be due to the enhanced migration and escape mechanisms from focal surgical or chemotherapeutic treatment.”

  1. Lines 198-201: The mechanisms of this statement must be explained and references presented.

This sentence is a continuation of the previous sentence’s reference. The following sentence is a  cumulative conclusion of the entire paragraph. We have added a reference to the end of the paragraph to demonstrate this.

  1. Lines 327-330: Which are the therapies that target osmotic and pressures imbalances?

We have deleted this sentence from the paper as it is a broad, unexplained statement. Discussing the mechanisms of treatment is not the focus of the section or paper, so the sentence did not add to the study.

  1. Chapter 4.2.1: Hypoxia isn’t a physical force.

We have removed hypoxia as a physical force from the manuscript.

  1. Chapter 4.2.2: Increased intracranial pressure is a clinical problem and is common complication of brain malignancies.

We recognize this and have added this to the discussion:

“Increased intracranial pressure is defined as an elevated pressure within the skull. This is a common clinical problem encountered in patients with brain tumors. However, the effects of increased pressure on tumor cell migration are not fully understood.”

  1. Line 382: Is the term postpartum correct?

Yes, this was a case report of a woman diagnosed with GBM in the postpartum period. We have rephrased this sentence for clarity:

“One case report of a woman diagnosed with GBM in the postpartum period without signs of myelopathy presented with ~20.6 mmHg opening pressure on lumbar puncture.”

  1. Chapter 425: Genetic mutations aren’t physical forces

We have removed “genetic mutations” from the table and discussion of the paper

Reviewer 2 Report

Abstract should be 200 words long. Currently has 206 words.

The article is a literature review from six months ago (September 13 th , 2021). The article must be updated with the latest articles.

The authors searched for articles with the phrase "... and (migration OR invasion)". So all the elaborate articles are influenced by the physical forces in GBM migration. If the article you were looking for was about, for example, proliferation but not migration, it was not found. Authors should change the title to "Physical Forces in GBM migraton" or search for all articles with the role of Physical Forces in GBM.

The article does not sufficiently discuss the molecular mechanisms of the influence of physical forces in GBM.

Incorrectly drawn figures. Hypoxia does not only cover one cell, but a much larger area.

The article is about Physical Forces in GBM. Why do the authors write about hypoxia?

Some of the abbreviations are not explained in the text.

Author Response

Abstract should be 200 words long. Currently has 206 words.

We thank this reviewer for their comments. We have shortened the abstract so that it is now 197 words.

The article is a literature review from six months ago (September 13 th , 2021). The article must be updated with the latest articles.

The search was updated through March 23, 2022. This identified 6 additional articles and 2 of these articles met inclusion criteria and were included in analysis.

The authors searched for articles with the phrase "... and (migration OR invasion)". So all the elaborate articles are influenced by the physical forces in GBM migration. If the article you were looking for was about, for example, proliferation but not migration, it was not found. Authors should change the title to "Physical Forces in GBM migraton" or search for all articles with the role of Physical Forces in GBM.

We have changed the title of the manuscript to reflect this.

The article does not sufficiently discuss the molecular mechanisms of the influence of physical forces in GBM.

We have included a paragraph at the end of the discussion which details the major molecular mechanisms associated with physical forces.

“4.3 Major Molecular Mechanisms Associated with Physical Forces in GBM

            We have reviewed 11 physical mechanisms involved in GBM aggression, recurrence, migration, and invasion and, in summary, we identified 34 influential molecules and pathways. These molecular influencers were elucidated primarily via analysis of human cell lines, and included G508, U373 MG, CD133+ GBM cells, U251, U87MG, U87, LN229, HGL21, U343, GL15, U118, and CC2565. U87 cell lines were used most frequently in the reviewed studies. Three molecular pathways (Piezo/PIEZO15, tenascin C6, Talin-17) were found to be involved in altering GBM stiffness, one (tenascin9) involved in tensile forces, one (PHIP10) involved in traction, four (miR54814, caveolin-1, integrin-β1, Rac115) involved in compression, two (PIVL16, P311/PTZ1717) involved in adhesion, ten (Swelling-activated chloride current22, caveolin-1/CAVIN1, UPA, MMPS, AQP123, Snail-1, Snail-2, N-cadherin, Twist, and vimentin24) involved in changes to cellular osmotic pressure, seven (Nestin, vimentin, actin filaments, vinculin, paxillin, and FAK25) contributing to shear stress, two (collagen, hyaluronan27) contributing to solid stress, and two (HAMLET28, swelling-activated chloride currents29) involved in cellular volume changes. Most of the contributing molecular changes were not directly overlapping, though several contributed to more than one physical force: tenascin contributed to both stiffness and tensile forces; swelling-activated chloride channels impacted both changes to cellular osmotic pressure and cellular volume changes; vimentin influenced osmotic pressure and shear stress; hyaluronan impacted solid stresses and genetic mutations. The role of multiple molecular pathways influencing single cellular functions highlights GBM redundancy and suggests that the physical forces associated with more molecular pathways are critical to GBM survival. Furthermore, the diversity and variability of molecular changes to GBMs are telling of GBM’s robustness and adaptability.

Incorrectly drawn figures. Hypoxia does not only cover one cell, but a much larger area.

We have removed the hypoxia discussion and image from the manuscript.

The article is about Physical Forces in GBM. Why do the authors write about hypoxia?

We have removed the hypoxia discussion and image from the manuscript.

Some of the abbreviations are not explained in the text.

Our apologies for this. We have gone through and defined all abbreviations.

Round 2

Reviewer 1 Report

Dear authors.

I agree with the corrections made.

Reviewer 2 Report

ok